# Mid-Pregnancy Fructosamine Measurement—Predictive Value for Gestational Diabetes and Association with Postpartum Glycemic Indices

**DOI:** 10.3390/nu10122003

**Published:** 2018-12-18

**Authors:** Véronique Gingras, Sheryl L. Rifas-Shiman, Karen M. Switkowski, Emily Oken, Marie-France Hivert

**Affiliations:** 1Division of Chronic Disease Research across the Lifecourse, Department of Population Medicine, Harvard Medical School and Harvard Pilgrim Health Care Institute, Boston, MA 02215, USA; veronique_gingras@harvardpilgrim.org (V.G.); sheryl_rifas@harvardpilgrim.org (S.L.R.-S.); karen_switkowski@harvardpilgrim.org (K.M.S.); emily_oken@harvardpilgrim.org (E.O.); 2Department of Nutrition, Harvard TH Chan School of Public Health, Boston, MA 02115, USA; 3Diabetes Unit, Massachusetts General Hospital, Boston, MA 02214, USA

**Keywords:** fructosamine, pregnancy, gestational diabetes, glucose tolerance, neonatal outcomes

## Abstract

Screening for gestational diabetes mellitus (GDM) during pregnancy is cumbersome. Measurement of plasma fructosamine may help simplify the first step of detecting GDM. We aimed to assess the predictive value of mid-pregnancy fructosamine for GDM, and its association with postpartum glycemic indices. Among 1488 women from Project Viva (mean ± SD: 32.1 ± 5.0 years old; pre-pregnancy body mass index 24.7 ± 5.3 kg/m^2^), we measured second trimester fructosamine and assessed gestational glucose tolerance with a 50 g glucose challenge test (GCT) followed, if abnormal, by a 100 g oral glucose tolerance test (OGTT). Approximately 3 years postpartum (median 3.2 years; SD 0.4 years), we measured maternal glycated hemoglobin (*n* = 450) and estimated insulin resistance (HOMA-IR; *n* = 132) from fasting blood samples. Higher glucose levels 1 h post 50 g GCT were associated with higher fructosamine levels (Pearson’s r = 0.06; *p* = 0.02). However, fructosamine ≥222 µmol/L (median) had a sensitivity of 54.8% and specificity of 48.6% to detect GDM (area under the receiver operating characteristic curve = 0.52); other fructosamine thresholds did not show better predictive characteristics. Fructosamine was also weakly associated with 3-year postpartum glycated hemoglobin (per 1 SD increment: adjusted β = 0.03 95% CI [0.00, 0.05] %) and HOMA-IR (per 1 SD increment: adjusted % difference 15.7, 95% CI [3.7, 29.0] %). Second trimester fructosamine is a poor predictor of gestational glucose tolerance and postpartum glycemic indices.

## 1. Introduction

Gestational diabetes mellitus (GDM) carries risk of adverse outcomes for the mother and child [1], with a linear relation between maternal glucose levels during pregnancy and risk of adverse outcomes [2]. Screening of all women in pregnancy between weeks 24 and 28 is currently recommended to ensure appropriate treatment of GDM and reduce associated risks [1,3]. For GDM screening, a two-step approach with a 50 g non-fasting glucose challenge test (GCT) with plasma glucose measured at 1 h, followed, if abnormal, with a 3-h 100 g fasting oral glucose tolerance test (OGTT) is currently endorsed by both the American Diabetes Association [4] and the American College of Obstetricians and Gynecologists [5].

GDM screening is essential to ensure appropriate care, yet the recommended approaches are burdensome for mothers and for healthcare personnel. Commonly done as a first step for GDM screening, the 50 g GCT is done over an hour and has limitations in terms of reproducibility and predictive value [6,7,8]. Previous studies have attempted to identify simpler markers of abnormal glucose tolerance in pregnancy, for example, using glycated circulating proteins, but yielded inconclusive or conflicting results. Glycated hemoglobin (hemoglobin A1c) is widely used for monitoring of blood glucose control among those with diabetes, and more recently, for diabetes diagnosis (outside of pregnancy) [4,9]. However, in pregnancy A1c levels are decreased due to the increased erythrocyte turnover [1,10]. In addition, A1c reflects glucose control over a longer period (up to 16 weeks), thus not fully reflecting changes in glycemia resulting from rapidly increasing insulin resistance that occurs in the second trimester of pregnancy. Fructosamine is a marker of glycemic control with the advantage that it reflects shorter term glucose levels (prior two–three weeks) given the relatively rapid turnover of circulating proteins [11] (in contrast to ~120 days of erythrocytes). Fructosamine is the result of glycosylation (covalent attachment of glucose or fructose) of any serum proteins, forming ketoamines [11]. In laboratory measurement, fructosamine represents all glycated proteins in circulation, with albumin being about 60% of the total serum proteins. Measurement of fructosamine is quick, inexpensive and technically simple.

Previous studies have assessed the ability of fructosamine to predict GDM and its association with neonatal outcomes [12,13,14,15] and have reported discordant conclusions about its usefulness. Overall, the predictive value of fructosamine for GDM and perinatal outcomes has been studied in small samples, in mostly homogenous populations and with inconsistent thresholds. Moreover, whether it is associated with postpartum glycemic indices has not been studied. We aimed to evaluate the predictive value of fructosamine in pregnancy for abnormal gestational glucose tolerance and to assess the associations of fructosamine with maternal postpartum glycemic indices. We also explored associations between fructosamine and neonatal outcomes in women with and without GDM. We hypothesized that elevated fructosamine levels would be associated with abnormal gestational glucose tolerance and with elevated postpartum glycemic indices.

## 2. Materials and Methods

We studied participants from Project Viva, a prospective pre-birth cohort recruited from offices of what is now Atrius Harvard Vanguard Medical Associates in Eastern Massachusetts, United States. Inclusion and exclusion criteria have been published [16] and are available in clinicaltrials.gov as NCT02820402. In addition, study questionnaires and instruments are available at www.hms.harvard.edu/viva/. The study was approved by the Institutional Review Board of Harvard Pilgrim Health Care. All participants provided written informed consent at enrollment and each postnatal follow-up visit. All procedures were in accordance with the ethical standards for human experimentation established by the Declaration of Helsinki.

From the 2128 mothers who had a live singleton birth and who were included in the cohort, we excluded women with missing fructosamine measurement (*n* = 540) or with measurement from samples with gross or moderate hemolysis or lipemia (*n* = 49), with pre-gestational type 1 or type 2 diabetes (*n* = 6), with missing glucose tolerance status in pregnancy (*n* = 21) and with missing covariates (*n* = 24). We thus included 1488 women with glucose tolerance assessment during pregnancy and fructosamine measurement. Compared to participants included in this analysis, those excluded (*n* = 640) were younger (31.1 vs. 32.1 years old), had a greater pre-pregnancy body mass index (BMI) (25.3 vs. 24.7 kg/m^2^), were less likely to be white (52 vs. 72%), to have a college degree (56 vs. 68%), to have an annual household income >USD 70,000 (55 vs. 63%) and to be married or cohabitating (88 vs. 93%). They were also more likely to be overweight or obese prior to pregnancy (41 vs. 36%) and to have developed gestational diabetes mellitus (7.6 vs. 4.9%). We found no differences between participants included vs. excluded for nulliparity, smoking in pregnancy and 50 g GCT results.

### 2.1. Exposure

We measured fructosamine from a blood sample collected between weeks 24 and 28. The blood samples were collected between 1999 and 2002, and samples were stored in liquid nitrogen. Fructosamine was measured in 2016–2017. Previous studies have suggested stability of this assay in frozen samples for short or long-term [17,18,19]. Fructosamine is a colorimetric assay which we performed using a Roche P Modular system (Roche Diagnostics, Indianapolis, IN, USA), using reagents and calibrators from Roche. The day-to-day variabilities at fructosamine concentrations of 288 and 512 µmol/L were 2.9% and 1.7%, respectively.

### 2.2. Outcomes

Gestational glucose tolerance was determined using a two-step approach. First, all women underwent routine clinical screening for GDM at 26 to 28 weeks using a random 50 g GCT with venous blood sampling 1 h post-load. Women with a blood glucose value below 140 mg/dL were considered as having normal glucose tolerance (NGT), whereas women with a blood glucose value ≥140 mg/dL were referred for a fasting 3-h 100-g OGTT. Abnormal OGTT values were defined as: >95 mg/dL at baseline, >180 mg/dL at 1 h, >155 mg/dL at 2 h and >140 mg/dL at 3 h. We categorized women with normal OGTT values but abnormal GCT as having isolated hyperglycemia (IH) and those with one abnormal OGTT value as having impaired glucose tolerance (IGT). Women with two or more abnormal OGTT values were clinically diagnosed as having gestational diabetes mellitus (GDM). Typically, GDM treatment would include diet therapy, daily fasting blood glucose monitoring and, in some cases, insulin therapy [20].

We obtained, from medical records, child’s weight recorded at birth by medical personnel, delivery date, and mode of delivery (cesarean or vaginal). We derived gestational age (GA) at birth using the date of the last menstrual period, or from a mid-pregnancy ultrasound if the two estimates differed by >10 days. Sex-specific birth weight for GA *z* scores were determined from US national reference [21]. Macrosomia was defined as birth weight above 4000 g, large for GA (LGA) was defined as birth weight for GA *z* score ≥90th percentile and small for GA (SGA) was defined as birth weight for GA *z* score <10th percentile.

At the early childhood visit from this longitudinal study (median child’s age 3.2 years, range 2.8 to 6.3 years), mothers were invited for a follow-up examination. Trained personnel measured mothers’ weight to the nearest 0.1 kg using an electronic scale (model 881; Seca Corp, Hanover, MD, USA) and height to the nearest 0.1 cm using a calibrated stadiometer, and we calculated BMI (kg/m^2^). A research phlebotomist collected blood specimens in the subsample of women who fasted for at least 12 h. Plasma fasting insulin was measured using a microparticle enzyme immunoassay on the IMZ analyzer (Abbott Laboratories, Chicago, IL, USA) and fasting glucose was measured enzymatically using the Hitachi 911 analyzer (Roche Diagnostics) with Roche Diagnostic reagents. Glycosylated hemoglobin (A1c) was assessed using the Hitachi 917 analyzer (Roche Diagnostics). We used the homeostatic model assessment to estimate insulin resistance (HOMA-IR: fasting glucose [mg/dL] × insulin [µU/L]/405) [22]. From the 1488 women included in this analysis, we excluded *n* = 86 women who were pregnant during the 3-year examination because these levels could be influenced by pregnancy, and 3-year postpartum fasting glucose, fasting insulin and HOMA-IR were available for 132 participants while A1c was available for 450 participants. Women included in the 3-year analysis had similar levels of fructosamine during pregnancy compared to women not included (mean (SD) = 232.3 (45.3) vs. 231.9 (46.0) µmol/L) as well as similar socio-demographic characteristics.

### 2.3. Covariates

At enrollment, women self-reported their age, education level, parity, household income, smoking status and history, race/ethnicity (non-Hispanic white, black, Hispanic, Asian, or other), height, and pre-pregnancy weight, from which we calculated pre-pregnancy BMI (kg/m^2^). For the current analyses, we dichotomized educational status as college graduate vs. not a college graduate, parity as nulliparous vs. multiparous, annual household income as > vs. ≤ USD 70,000, and pregnancy smoking status as yes vs. no (including former and never smokers), and pre-pregnancy BMI as overweight/obese (≥ 25 kg/m^2^) vs. not overweight/obese. We calculated first trimester weight gain as the weight gain during the first 91 days of pregnancy using self-reported pre-pregnancy weight and clinical weights extracted from medical records.

### 2.4. Statistical Analysis

We presented maternal characteristics according to quantiles of second trimester fructosamine as mean ± SD or number (%). Differences in characteristics of participants according to fructosamine levels were tested for statistical significance using unadjusted linear regression models.

We examined associations of fructosamine with GCT results (per 1 SD increment of 1-h glucose; 26.3 mg/dL) using linear regression models. We also examined the association of plasma fructosamine with GCT results using Pearson correlation.

We evaluated the sensitivity and specificity of fructosamine levels ≥ the 50th percentile (222 µmol/L), ≥ the 75th percentile (256 µmol/L) and ≥ the 95th percentile (312 µmol/L) to detect GDM and abnormal OGTT results (IGT and GDM). We generated receiver operating characteristic (ROC) curves to examine the predictive value of fructosamine for GDM, macrosomia, cesarean section delivery, LGA, SGA and preterm birth (< 37 weeks of gestation).

We assessed associations of fructosamine (per 1 SD increment; 45.8 µmol/L) as well as 50 g GCT results (per 1 SD increment) with birth weight and birth weight for GA *z* score in women with and without GDM using multivariable linear regression models. We performed sensitivity analyses adjusting the models for maternal race/ethnicity, age, parity, smoking status in pregnancy, pre-pregnancy BMI and first trimester weight gain. We decided a priori to conduct analyses separately for mothers with or without GDM in pregnancy since women with GDM typically receive lifestyle counselling (±insulin treatment), which has been shown to influence neonatal outcomes.

We also assessed associations of fructosamine and GCT results with maternal glycemic indices at 3 years postpartum (fasting glucose and insulin, A1c and HOMA-IR) using multivariable linear regression models and sensitivity analyses were performed adjusting for the same covariates as for birth weight for GA *z* score. We log-transformed maternal HOMA-IR and fasting insulin at 3 years postpartum since they were right-skewed, and we present regression coefficients and confidence intervals for these outcomes as percentage of difference after exponentiation.

To address the issue of missing baseline fructosamine measurements in 589 women, we implemented inverse probability weighting (IPW) as a sensitivity analysis. First, we predicted the probability of missing fructosamine, based on the following covariates (maternal age, pre-pregnancy BMI, race/ethnicity, education, marital status, parity, smoking during pregnancy, first trimester weight gain and household income). Next, we re-ran our main analysis weighted by the inverse of the probability of having fructosamine measured. Results were similar after implementing IPW methods (Appendix A). We conducted all analyses using SAS version 9.4 (SAS Institute, Cary, NC, USA).

## 3. Results

Maternal and household characteristics are presented in Table 1. Participants included in this analysis were (mean ± SD) 32.1 ± 5.0 years old and had a pre-pregnancy BMI of 24.7 ± 5.3 kg/m^2^, a fructosamine level of 232.0 ± 45.8 µmol/L, and a GCT result of 114.3 ± 26.3 mg/dL. They were mostly white (72%), college educated (68%) and non-smokers (88%) and they had an annual household income >USD 70,000 (63%). Participants who smoked during pregnancy (β = −8.3 vs. never or former smokers, 95% CI [−15.5, −1.1] µmol/L) and with a higher maternal pre-pregnancy BMI (per 1 kg/m^2^; β = −0.7, 95% CI [−1.1, −0.2] µmol/L) had lower fructosamine levels. No associations of fructosamine were found with other socio-demographic characteristics.

We found a weakly positive correlation between GCT result and fructosamine (Pearson’s r = 0.06; *p* = 0.02). In addition, higher fructosamine levels were observed in women with higher 1 h-glucose values after GCT (per 1 SD increment of 1 h-glucose; β = 2.7 95% CI [0.3, 5.0] µmol/L), with abnormal GCT result (> vs. ≤140 mg/dL; β = 5.5 [−0.8, 11.7] µmol/L) and with a GDM diagnosis vs. IGT/IH/normal glycemia (β = 11.6 [0.9, 22.4] µmol/L). Adjustments for socio-demographic characteristics, first trimester gestational weight gain and pre-pregnancy BMI slightly strengthened the effect estimates (e.g., for GCT result > vs. ≤140 mg/dL: β = 7.1 [0.7, 13.5] µmol/L and for GDM diagnosis vs. IGT/IH/normal glycemia: β = 14.7 [3.7, 25.6] µmol/L). Results were very similar after implementing IPW (Appendix A).

Sensitivity and specificity of fructosamine to detect abnormal gestational glucose tolerance was then assessed using the three different fructosamine cut-points (Table 2). The sensitivity to detect GDM was 54.8% with a fructosamine level above the 50th percentile (222 µmol/L) with a specificity of 48.6%. As expected, sensitivity was lower with a fructosamine level above the 75th percentile (256 µmol/L; 26.0%) and above the 95th percentile (312 µmol/L; 6.9%), while specificity was higher (74.9% and 95.1%, respectively). Sensitivity and specificity of fructosamine to detect IGT and GDM together were similar to those for GDM alone.

Among participants included in this study, 73 (4.9%) had a diagnosis of GDM, 332 (22.3%) had a cesarean section delivery, and 91 participants (6.1%) had a preterm delivery. Among newborns of participants, 239 (16.1%) were characterized with macrosomia, 82 (5.5%) were SGA and 205 (13.8%) were LGA. ROC curves showed poor predictive value of fructosamine for GDM (area under the ROC curve (AUC) = 0.52), macrosomia (AUC = 0.48 in GDM women and 0.52 in non-GDM women), LGA (AUC = 0.59 in GDM women and 0.53 in non-GDM women), SGA (AUC = 0.57 in GDM women and 0.51 in non-GDM women), preterm birth (AUC = 0.46 in GDM women and 0.50 in non-GDM women) or cesarean section delivery (AUC = 0.58 in GDM women and 0.54 in non-GDM women). Results were almost identical after implementing IPW (Appendix A).

In addition, we did not find associations of mid-pregnancy fructosamine (per 1 SD increment) with birth weight (overall mean 3499.9; SD 535.2 g) in newborns from non-GDM mothers (unadjusted β = 4.1, 95% CI [−25.1, 33.2] g) and GDM mothers (unadjusted β = 26.0, 95% CI [−56.7, 108.6] g). Associations remained similar or slightly stronger, with confidence intervals crossing the null when adjusted for socio-demographic characteristics, first trimester gestational weight gain and pre-pregnancy BMI. Associations were also non-significant for birthweight for GA *z* score (overall mean 0.20; SD 0.96). Higher glucose levels during the GCT (per 1 SD increment) were associated with higher birth weight (unadjusted β = 72.9 [42.6, 103.3] g) and birthweight for GA *z* score (unadjusted β = 0.15 [0.09, 0.20] units) in women without GDM. The effect estimate was slightly attenuated (adjusted β = 54.2 [24.2, 84.1] g and 0.11 [0.05, 0.16] units for birth weight and birthweight for GA *z* score, respectively) after adjustment for socio-demographic characteristics, first trimester gestational weight gain and pre-pregnancy BMI. We did not find significant associations between GCT results and birth weight in GDM women (unadjusted β = 29.9, 95% CI [−145.8, 205.6] g). Results were very similar after implementing IPW (Appendix A).

Finally, we examined associations of second trimester fructosamine levels with 3-year postpartum glycemic indices (Table 3). Mean (SD) maternal 3-year postpartum A1c was 5.1 (0.3) %, fasting insulin was 10.3 (8.3) µU/L, fasting glucose was 74.9 (15.7) mg/dL and HOMA-IR was 1.97 (2.10). Higher fructosamine levels during pregnancy (per 1 SD increment) were associated with higher maternal 3-year postpartum A1c (unadjusted β = 0.02, 95% CI [0.00, 0.05] %), fasting insulin (unadjusted % difference = 15.2 [3.0, 28.9] %) and HOMA-IR (unadjusted % difference = 18.9 [5.1, 34.5] %). Higher GCT results during pregnancy (per 1 SD increment of blood glucose) were also associated with higher A1c (unadjusted β = 0.05, 95% CI [0.03, 0.08] %), fasting insulin (unadjusted % difference = 16.6 [4.8, 29.7] %) and HOMA-IR (unadjusted % difference = 23.3 [9.9, 38.4] %), and additionally associated with higher fasting glucose (unadjusted β = 3.8 [1.2, 6.3] mg/dL). All associations were slightly attenuated when adjusted for socio-demographic characteristics, parity, smoking status in pregnancy, first trimester gestational weight gain and pre-pregnancy BMI. Results were very similar after implementing IPW (Appendix A).

## 4. Discussion

We showed that higher second-trimester fructosamine levels were associated with abnormal GCT results and GDM, albeit with small effect sizes. However, fructosamine had a poor sensitivity to detect abnormal gestational glucose tolerance, with no acceptable cut-point identified. Moreover, fructosamine did not predict neonatal outcomes, including macrosomia, LGA and SGA. Finally, higher fructosamine in pregnancy was also minimally associated with greater insulin resistance as estimated by HOMA-IR measured in women 3-year postpartum. Overall, our findings suggest that fructosamine measured in the second trimester does not show adequate predictive value for identification of GDM women and is only weakly associated with postpartum glycemic indices.

Since the discovery of fructosamine nearly 40 years ago, several studies have investigated the diagnostic value of this test for abnormal gestational glucose tolerance. The first studies in the 1980s showed a good sensitivity (85%) of fructosamine measured at 29 weeks’ gestation in 99 women (including 19 women with pre-gestational diabetes) to detect GDM [23] and a positive association between first-trimester fructosamine and birth weight ratio in 30 women with pre-gestational diabetes [24]. However, subsequent studies showed a poor sensitivity of fructosamine for abnormal gestational glucose tolerance [12,13,15,25,26,27], ranging from 0 to 86% depending on the cut-points used (between 230 and 270 µmol/L), and studies that investigated neonatal outcomes (birth weight, macrosomia and neonatal morbidity) did not find significant associations [13,14,28]. Most of these studies had limited sample sizes (<200 participants) [13,15,25,27] or did not include neonatal outcomes [12,15,25,26,27]. In contrast, our analysis included over 1400 women, and had numerous outcomes, including glycemic screening, birth outcomes, and maternal postpartum outcomes. Our results are consistent with the most recent studies showing a poor sensitivity of fructosamine to detect impaired gestational glucose tolerance and GDM. Similar to our findings, previous studies showed an association between fructosamine and GCT results and showed elevated fructosamine levels at the time of GDM diagnosis [29]. Yet, in the present study, effect estimates were modest, and observed associations do not translate into adequate detection of GDM or gestational glucose intolerance.

In non-pregnant populations, fructosamine has been shown to adequately identify individuals with diabetes, and to improve diabetes detection when used in combination with A1c or fasting glucose [30]. In the Atherosclerosis Risk in Communities (ARIC) study (*n* > 11,000 adults), fructosamine > 95th percentile (264 µmol/L) was associated with incident diabetes (HR 4.96, 95% CI [4.36, 5.64]) [31]. In the AMORIS cohort (*n* = 10,987), fructosamine was strongly correlated with fasting glucose and A1c, and a fructosamine level of 250 µmol/L had a sensitivity of 61% and a specificity of 97% to detect type 2 diabetes [32]. In contrast, most studies, including the present analysis, showed poor predictive value of fructosamine to identify impaired glucose tolerance in pregnant women. Fructosamine levels are influenced by conditions affecting serum albumin levels or relative hypo-proteinemia, which can be seen in pregnancy [33]. It is possible that direct measurement of glycated albumin (not affected by total serum albumin levels) would have resulted in better predicting abilities. In addition, studies showed variations of fructosamine with gestational age [33], which complicates the identification of possible cut-points throughout pregnancy.

Identification of maternal impaired glucose tolerance is crucial to reduce risks associated with maternal hyperglycemia [1,2]; however, it remains costly and time consuming. It is possible that identifying simpler methods for GDM screening could further improve and facilitate GDM screening rates. Moreover, even the 50 g GCT recommended in the two-step approach has limited predictive value for gestational glucose tolerance [6,26]. Benhalima et al. recently reported a 59.6% sensitivity to detect GDM using the 140 mg/dL threshold from the GCT [6]. Perhaps prediction or detection algorithms for diabetes in pregnancy could be derived using simpler markers of glycemic control concomitantly. For example, blood lipids have been shown to be elevated while adiponectin levels have been shown to be lower in the weeks prior to a diagnosis of GDM [15,29]. These biomarkers, alone or in combination with traditional risk factors, could potentially allow to alleviate some of the burden associated with GDM screening, and should be further investigated.

This is, to our knowledge, the largest study examining the predictive value of fructosamine for impaired gestational glucose tolerance and neonatal outcomes in women with and without GDM. Our study adds to previous literature by assessing multiple neonatal outcomes, and maternal outcomes in the postpartum years. Limitations include the single measurement of fructosamine in the second trimester of pregnancy; it would have been informative to have measurements of fructosamine and/or HbA1c earlier in pregnancy or prior to conception. Although insulin resistance typically appears during the second trimester of pregnancy, it is possible that a single fructosamine measurement during this period did not fully capture recent changes in insulin resistance. Also, our study sample mainly included women of relatively high socio-economic status, limiting generalizability. We excluded women who were currently pregnant from our 3-year postpartum analysis, perhaps also limiting generalizability. Finally, the GCT was performed in non-fasting conditions and it remains possible that fructosamine would be better correlated with fasting glucose measurements or with other measures of glycemic variability.

## 5. Conclusions

In summary, we found weak associations between second trimester glucose values from the GCT and fructosamine, and fructosamine was a poor predictor of abnormal gestational glucose tolerance. In addition, fructosamine levels were not associated with birth size and were weakly associated with maternal 3-year postpartum glycemic indices. To conclude, plasma fructosamine did not show adequate predictive characteristics for detecting impaired glucose tolerance in pregnancy or for predicting postpartum glycemic status.

## Figures and Tables

**Table 1 nutrients-10-02003-t001:** Sociodemographic and weight-related characteristics according to second trimester fructosamine in *N* = 1488 women from Project Viva.

	Fructosamine, µmol/L					
	<180 (5th percentile) *N* = 69	≥180, <200 (5–25th percentile) *N* = 286	≥200, <222 (25–50th percentile) *N* = 366	≥222, <256 (50–75th percentile) *N* = 393	≥256, <312 (75–95th percentile) *N* = 299	≥312 (95th percentile) *N* = 75
Age, years	31.6 ± 4.9	32.0 ± 5.0	32.1 ± 4.7	32.3 ± 5.2	32.2 ± 5.0	32.5 ± 5.4
Race/ethnicity, *n* (%) ^1^						
White	51 (74)	193 (67)	267 (73)	289 (74)	221 (74)	55 (73)
Black	7 (10)	44 (15)	54 (15)	56 (14)	46 (15)	6 (8)
Hispanic	5 (7)	22 (8)	14 (4)	26 (7)	16 (5)	3 (4)
Asian	1 (1)	18 (6)	22 (6)	8 (2)	10 (3)	6 (8)
Other	5 (7)	9 (3)	9 (2)	14 (4)	6 (2)	5 (7)
Household income > USD 70,000 /year, *n* (%)	43 (70)	153 (58)	223 (66)	232 (65)	165 (60)	50 (72)
Education ≥ College degree, *n* (%)	45 (65)	182 (64)	273 (75)	257 (65)	204 (68)	52 (69)
Marital status, married/cohabitating, *n* (%)	65 (94)	263 (92)	344 (94)	359 (91)	282 (94)	70 (93)
Nulliparous, *n* (%)	37 (54)	119 (42)	179 (49)	202 (51)	149 (50)	37 (49)
Smoking during pregnancy, yes, *n* (%)	7 (10)	42 (15)	50 (14)	42 (11)	31 (10)	5 (7)
Pre-pregnancy body mass index, kg/m^2^	26.1 ± 5.2	25.6 ± 5.9	24.6 ± 5.3	24.6 ± 5.3	24.0 ± 4.6	23.9 ± 4.6
Pre-pregnancy overweight/obesity, *n* (%)	34 (49)	120 (42)	126 (34)	130 (33)	97 (32)	24 (32)
First trimester weight gain, kg	2.7 ± 3.3	2.9 ± 2.9	2.8 ± 2.6	3.1 ± 3.0	2.6 ± 2.5	2.7 ± 3.1

^1^ Due to rounding issues, not all column percentages add up to 100%.

**Table 2 nutrients-10-02003-t002:** Sensitivity and specificity of second trimester fructosamine to detect abnormal glucose tolerance during pregnancy in *N* = 1488 women from Project Viva.

		Abnormal OGTT Result: IGT and GDM	Abnormal OGTT Result: GDM
Fructosamine ≥ 222 µmol/L (≥50th percentile)	Sensitivity	51.3	54.8
	Specificity	48.4	48.6
	Positive predictive value	8.0	5.2
	Negative predictive value	92.0	95.4
Fructosamine ≥ 256 µmol/L (≥75th percentile)	Sensitivity	25.2	26.0
	Specificity	74.9	74.9
	Positive predictive value	8.0	5.1
	Negative predictive value	92.0	95.2
Fructosamine ≥ 312 µmol/L (≥95th percentile)	Sensitivity	8.4	6.9
	Specificity	95.3	95.1
	Positive predictive value	13.3	6.7
	Negative predictive value	92.3	95.2

OGTT: oral glucose tolerance test; IGT: impaired glucose tolerance; GDM: gestational diabetes mellitus.

**Table 3 nutrients-10-02003-t003:** Multivariable associations of second trimester fructosamine with maternal 3-year postpartum glycemic indices (*n* = 451 ^1^).

	Fructosamine, per 1 SD Increment		GCT Result, per 1 SD Increment	
	Unadjusted	Adjusted ^2^	Unadjusted	Adjusted ^2^
	**β [95% CI]**			
A1c, %	0.02 [0.00, 0.05]	0.03 [0.00, 0.05]	0.05 [0.03, 0.08]	0.04 [0.02, 0.07]
Fasting glucose, mg/dL	2.8 [0.1, 5.5]	2.7 [−0.1, 5.5]	3.8 [1.2, 6.3]	3.6 [0.8, 6.4]
	**% difference [95% CI]**			
Log fasting insulin	15.2 [3.0, 28.9]	12.3 [2.1, 23.5]	16.6 [4.8, 29.7]	11.4 [1.1, 22.7]
Log HOMA-IR	18.9 [5.1, 34.5]	15.7 [3.7, 29.0]	23.3 [9.9, 38.4]	17.6 [5.4, 31.2]

^1^*N* = 132 for fasting glucose, fasting insulin and HOMA-IR and *N* = 450 for A1c; ^2^ Adjusted for maternal age, race/ethnicity, parity, smoking during pregnancy, first trimester weight gain and maternal pre-pregnancy BMI; GCT: glucose challenge test; HOMA-IR: homeostatic model assessment for insulin resistance.

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
