# Peer review of "Mid-Pregnancy Fructosamine Measurement—Predictive Value for Gestational Diabetes and Association with Postpartum Glycemic Indices"

_nutrients, 2018, doi:10.3390/nu10122003_

Reviewer 1 Report

Dear authors,

Thank you for the opportunity to read your manuscript, "Mid-pregnancy fructosamine measurement - predictive value for gestational diabetes and association with postpartum glycemic indices". I thnk this is very important work.

I have a number of minor (mostly editorial) concerns and a few major concerns that I would appreciate you addressing. 

Minor:

line 17: the number "2" needs to exponentiated

line 22 (and elsewhere): Throughout the article "Pearson R" is referred to. To my understanding the correct spelling an puncuation is "Pearson's r" (possessive and lower case)

line 33: change relationship to relation

line 52: as a matter of word choice, as protein itself is not alive it should not be considered as having a "lifespan" 

lines 108-111 (and in many places): in many places inequalities are used. their use should be consistent (Are they flush against the number or is there a space in between the two.

line 173 (and in many places): refer to beta estimates using and "=" rather than just stating beta and a number.

line 177: I would recommend capitalizing "L" for ease in reading

Table 2: N (the whole study sample) vs. n (a portion of the study sample)

Page numbers are always not correct.

Major:

lines 50-52: Given the importance of Fructosamine to the manuscript, the authors should elaborate further on what is it.

lines 73-84: The amount of missingness (~25%) is deeply concerning. It also appears that those that are missing are from a lower SES background. The racial background difference mentioned in line 80 is also very concerning (20% difference). This needs to be addressed further. Consider handling missingness through multiple imputation, inverse probabilty weighting, or both.

line 123: Using this exclusion could you be excluding those that are more fertile? The consequences of this exclusion needs to be addressed

line 147: Why is a<34 weeks gestation cut-off used? Why not <32 weeks (very preterm) or <37 weeks (preterm)?

Table 3: Are Fructosamine and GCT Result normally distributed?

I look forward to seeing this manuscript progress. Thank you for your time and attention to these concerns.

Best,

A.R.

Reviewer 2 Report

The work by Gingras et al is addressing a relevant issue, i.e. how to trust on some biomarkers such as fructosamine in gestational diabetes to use it as predictive vale of possible gestation  complications. The experimental work is novel, the statistical analysis is rigorous and the work is certainly of interest to the readers of Nutrients. I have only some minor comments:

 1.    The data presented in Table 1 are not complete, because the total of the race/ethnicity does not make 100 % always. Indeed, only in the column between 200 and 222 mmol/l the total is 100 %.

2.    What the HbA1cvalue was before pregnancy? It would be very useful to have such information, if available. Please comment such issue.

3.    The authors have to discuss some possible limitation of the fructosamine assay respect to the determination of glycated albumin, now currently available and much more specific. This could be a limitation of this work. Please discuss this point.

Author Response

Round  2

Reviewer 1 Report

Dear authors,

Thank you responding so positively to my earlier comments. I recognize that they were not the easiest concerns to address nor the quickest.

I have a few more questions/concerns/comments:

In the updated article you state that the IPW results are virtually identical but they appear to be far from identical. I would recommend including them in supplemental tables and describing how they vary any to what degree. 

For the inequalities listed at the top of Table 1, I would recommend put a comma between separate inequalities (e.g. "≥ 180, < 200")

In my earlier peer-review of the manuscript I stated, "line 123: Using this exclusion could you be excluding those that are more fertile? The consequences of this exclusion needs to be addressed." Your response is given below. Do you think that the women you are excluding would be more fertile on average than the women who were not pregnant at the 3-y follow-up visit? I recognize that pregnancy depends upon a whole host of other factors in addition to fertility, but I think it may be worth mentioning in your discussion and indicating whether or not this exclusion makes results less generalizable. It is reassuring to know that among measured factors the excluded and non-exlcuded women have comparable average characteristics.

     We did not measure glucose, insulin, and A1c values for 86 women who were pregnant at the 3-year follow-up visit because these levels could be influenced by pregnancy. We did not exclude women who had pregnancies after the index pregnancy, but who were not currently pregnant at the 3-year follow-up visit. We compared characteristics of women included vs non-included in this analysis and have amended the text (lines 127-132):

     From the 1488 women included in this analysis, we excluded n = 86 women who were pregnant during the 3-y examination because these levels could be influenced by pregnancy, and 3-year postpartum fasting glucose, fasting insulin and HOMA-IR were available for 132 participants while A1c was available for 450 participants. Women included in the 3-y analysis had similar levels of fructosamine during pregnancy compared to women not included (mean (SD) = 232.3 (45.3) vs 231.9 (46.0) µmol/L) as well as similar socio-demographic characteristics.

I greatly appreciate your response. I will recommend to the editor that only minor revisions are suggested.

Best,

A.R.

Author Response

Dear Editor and reviewers,

 Thank you for giving us the opportunity to revise and resubmit our paper.

We appreciate the thoughtful reviews, which we believe have resulted in a substantially clearer presentation of our results.

We have responded to your comments, with line numbers referring to the track-changed version. We highlighted the new changes for Revision #2 in yellow.

We hope that our responses allow for publication, but we are glad to continue to work on this paper if any additional revisions are needed.

Dear authors,

Thank you responding so positively to my earlier comments. I recognize that they were not the easiest concerns to address nor the quickest. I have a few more questions/concerns/comments:

In the updated article you state that the IPW results are virtually identical but they appear to be far from identical. I would recommend including them in supplemental tables and describing how they vary any to what degree. 

Thank you for this suggestion. We have now included the IPW results as Supplemental Tables 1-4. Please see the end of this document. We also uploaded the supplemental tables as a separate file.

Lines 173-174: Results were similar after implementing IPW methods (Supplementary Tables).

Lines 191-192: Results were very similar after implementing IPW (Supplemental Table 1).

Lines 214: Results were almost identical after implementing IPW (Supplemental Table 2).

Lines 227-228: Results were very similar after implementing IPW (Supplemental Table 3).

 Lines 240-241: Results were very similar after implementing IPW (Supplemental Table 4).

For the inequalities listed at the top of Table 1, I would recommend put a comma between separate inequalities (e.g. "≥ 180, < 200")

We have now put a comma between separate inequalities (top of Table 1).

In my earlier peer-review of the manuscript I stated, "line 123: Using this exclusion could you be excluding those that are more fertile? The consequences of this exclusion needs to be addressed." Your response is given below. Do you think that the women you are excluding would be more fertile on average than the women who were not pregnant at the 3-y follow-up visit? I recognize that pregnancy depends upon a whole host of other factors in addition to fertility, but I think it may be worth mentioning in your discussion and indicating whether or not this exclusion makes results less generalizable. It is reassuring to know that among measured factors the excluded and non-exlcuded women have comparable average characteristics.

We did not measure glucose, insulin, and A1c values for 86 women who were pregnant at the 3-year follow-up visit because these levels could be influenced by pregnancy. We did not exclude women who had pregnancies after the index pregnancy, but who were not currently pregnant at the 3-year follow-up visit. We compared characteristics of women included vs non-included in this analysis and have amended the text (lines 127-132):

From the 1488 women included in this analysis, we excluded n = 86 women who were pregnant during the 3-y examination because these levels could be influenced by pregnancy, and 3-year postpartum fasting glucose, fasting insulin and HOMA-IR were available for 132 participants while A1c was available for 450 participants. Women included in the 3-y analysis had similar levels of fructosamine during pregnancy compared to women not included (mean (SD) = 232.3 (45.3) vs 231.9 (46.0) µmol/L) as well as similar socio-demographic characteristics.

Thank you for this point. We have now added a sentence to the limitations (lines 305-306):

We excluded women who were currently pregnant from our 3-year postpartum analysis, perhaps also limiting generalizability.

I greatly appreciate your response. I will recommend to the editor that only minor revisions are suggested.

Best,

A.R.